# TRACKING ANY POINT IN MULTI-VIEW VIDEOS

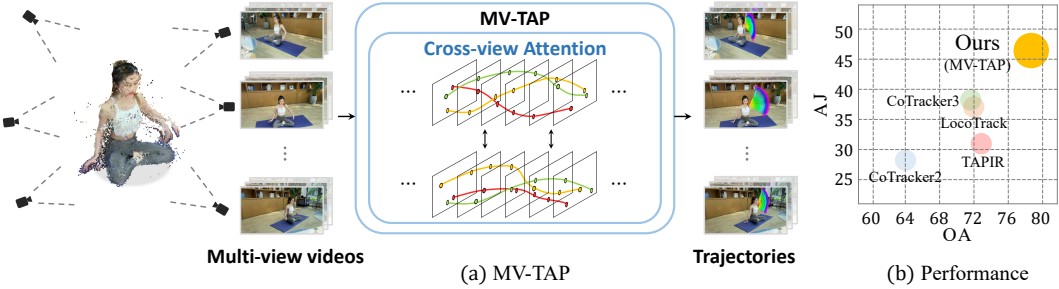

**Figure 1: Teaser.** We present **MV-TAP**, the first model for the new task of multi-view point tracking. Unlike dominant single-view point tracking methods (Karaev et al., 2024b; Doersch et al., 2023), which often degrade under high dynamics due to missing geometric cues in monocular video, our method aggregates complementary information from multi-view inputs, substantially outperforming single-view baselines.

## ABSTRACT

Accurate point tracking across video frames is a core challenge in computer vision, but existing single-view approaches often fail in dynamic real-world settings due to the limited geometric information in monocular video. While multi-view inputs provide complementary geometric cues, most current correspondence methods assume rigid scenes, calibrated cameras, or other priors that are rarely available in casual captures. In this work, we introduce the task of multi-view point tracking, which seeks to robustly track query points across multiple, uncalibrated videos of dynamic scenes. We present MV-TAP, a framework that leverages cross-view attention to aggregate spatio-temporal information across views, enabling more complete and reliable trajectory estimation. To support this new task, we construct a large-scale synthetic dataset tailored for multi-view tracking. Extensive experiments demonstrate that MV-TAP outperforms single-view tracking methods on challenging benchmarks, establishing an effective baseline for advancing multi-view point tracking research.

## 1 INTRODUCTION

Accurately tracking points across video frames, a task known as point tracking (Harley et al., 2022; Doersch et al., 2022), is a fundamental task in computer vision. It underpins a vast array of applications, including embodied AI (Vecerik et al., 2024; Bharadhwaj et al., 2024), autonomous driving (Balasingam et al., 2024), 4D reconstruction (Wang et al., 2024; Feng et al., 2025), and video editing (Geng et al., 2025; Jeong et al., 2025). Despite significant progress, the performance of existing *single-view* point tracking models often degrades in complex real-world scenarios.

The challenge often stems from the inherent ambiguity of using a single viewpoint. In this monocular setting, single-view point tracking models struggle with frequent occlusions, erratic motion, and depth uncertainty, as a single 2D projection point provides insufficient geometric constraints to resolve these ambiguities. Consequently, for downstream tasks demanding precise geometric consistency, the trajectories produced by even state-of-the-art trackers can be unreliable.

While multi-view systems offer a promising way to add these geometric constraints, existing multi-view correspondence methods are often ill-suited for this problem. Many of these techniques are designed for static scenes (Schonberger & Frahm, 2016), assume rigid geometry, and require known camera parameters, depth, or other geometric priors (Zhang et al., 2025a) that are unavailable in the casual, in-the-wild video captures where robust point tracking is most needed. This leaves a critical gap in methodology: there is no established paradigm for leveraging *multiple, uncalibrated videos* to track 2D points through dynamic scenes, even though such a capability would directly benefit many point tracking applications in scenarios where multi-view data is available.

Our key insight is that jointly processing multiple, uncalibrated video streams of a dynamic scene provides useful spatio-temporal constraints that can help resolve the ambiguities in single-view tracking. For instance, a point that is occluded or motion-blurred in one view may be clearly visible in other views. By reasoning across these views simultaneously, we can enforce spatio-temporal consistency and recover a more robust and accurate representation of the point's trajectory.

To this end, we formulate the task of multi-view point tracking, which aims to track a set of query points throughout multiple, uncalibrated videos of a dynamic scene. We also present MV-TAP (Tracking Any Point in Multi-view Videos), a framework that uses a cross-view attention mechanism to effectively aggregate information across all views and timesteps, building a holistic understanding of the dynamic scene. Furthermore, to facilitate research in this area, we construct a large-scale synthetic dataset specifically designed for training multi-view point tracking models. Our experiments on multi-view benchmarks show that this approach improves upon current tracking methods, leading to more complete and accurate predictions. Our code and dataset will be made publicly available.

Our contributions are summarized as follows:

- We define, for the first time, the task of multi-view point tracking for establishing robust spatio-temporal correspondences in dynamic scenes using multiple, uncalibrated videos.

- We propose MV-TAP, a framework that leverages cross-view information to address inherent limitations of single-view point tracking, such as occlusion and motion ambiguity.

- We demonstrate through extensive experiments that our method achieves competitive performance, providing an effective baseline for this new task.

## 2 RELATED WORK

**Point Tracking.** Point tracking aims to predict trajectories and occlusion states of given query points over time in a monocular video. PIPs (Harley et al., 2022) constructs local correlation volumes for initial point localization and iteratively refines it to obtain trajectory. Building upon this approach, TAPIR (Doersch et al., 2023) injects the idea of TAP-Net (Doersch et al., 2022) which construct per-frame global correlation with iterative refiner. CoTracker (Karaev et al., 2024b) introduces transformer-based refinement over 2D correlation features, and LocoTrack (Cho et al., 2024) further enhances robustness by constructing bidirectional local 4D correlation volumes to establish more robust tracking. Chrono (Kim et al., 2025a) extends pre-trained DINOv2 backbone with additional temporal adapter, yielding robust feature representation that captures long-term temporal context for accurate point tracking. Recent efforts have focused on addressing the reliance on synthetic data for training. BootsTAP (Doersch et al., 2024), CoTracker3 (Karaev et al., 2024a), and AnthroTAP (Kim et al., 2025b) aim to reduce the sim-to-real gap by leveraging real videos through self-supervised or pseudo-labeling strategies. Recently, TAP-Next (Zholus et al., 2025) reformulates point tracking as a next-token prediction problem, while AllTracker (Harley et al., 2025) leverages optical flow to incorporate temporal priors and achieve strong performance even at high resolution.

In this work, we introduce a new task of multi-view point tracking that analyzes synchronized videos from multiple cameras to exploit cross-view information, enabling more accurate trajectory estimation and robust occlusion handling.

**Multi-view Matching.** Classical multi-view matching predicts correspondences across unordered images, typically coupling local features with geometric consistency. SIFT (Lowe, 2004) introduced robust scale and rotation-invariant descriptors for keypoint matching. On this foundation, Agarwal et al. (2011) scaled SfM via correspondence estimation through pairwise geometric verification and

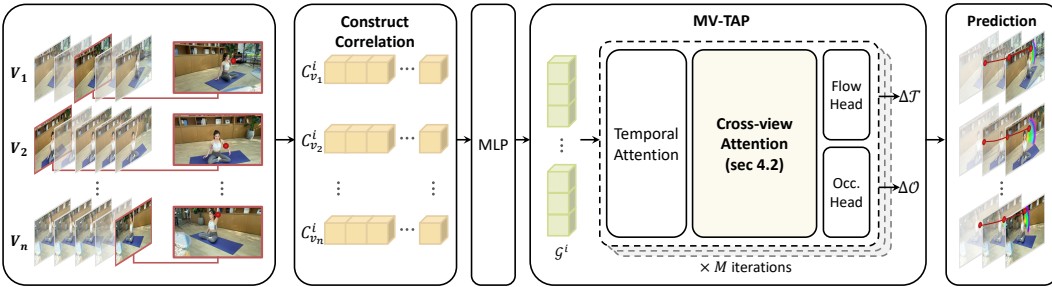

**Figure 2: Overall architecture of MV-TAP.** Given synchronized multi-view videos, per-view correlation volumes are extracted from a CNN encoder feature for each query point. These correlations are then fed into tracking head and iteratively update tracks and occlusion states. Within the tracking head, the inputs from multiple views are aggregated through cross-view attention module.

view-graph clustering. Sattler et al. (2017) showed that accurate visual localization is possible without a full 3D map by using retrieval-based matching and 2D–3D consistency across views. Recent learning-based approaches have improved this task: End2End (Roessle & Nießner, 2023) jointly predicts correspondences and poses with a differentiable framework. CoMatcher (Zhang et al., 2025b) leverages cross-view projection constraints and covisibility cues to establish globally consistent multi-view tracks from noisy pairwise matches. CER-MVS (Ma et al., 2022) demonstrates how cross-view correspondence can be integrated with cost volume aggregation for accurate depth estimation. However, these methods optimize for spatial reconstruction or per-frame matching without maintaining temporal consistency across video sequences.

In this work, we formalize *multi-view point tracking* in synchronized videos as recovering cross-view, temporally continuous 2D trajectories and visibilities from sparse query points, without 3D reconstruction or known camera calibration.

## 3 TASK DEFINITION

Previous point tracking methods (Harley et al., 2022; Doersch et al., 2022) on monocular videos often suffer from inherent ambiguity due to their restricted geometric information. Multi-view videos provide geometric cues which can help to address the ambiguities. Motivated by this, we introduce **multi-view point tracking** task as a new problem setting, which aims to construct robust spatio-temporal correspondences across views.

Formally, we define this task as follows. The inputs are multi-view videos $\mathcal{V} \in \mathbb{R}^{V \times T \times H \times W \times 3}$ and query points $\mathcal{Q} \in \mathbb{R}^{V \times N \times 3}$, where $V$ denotes the number of multi-view videos, $T$ represents the number of frames, and each frame has spatial resolution $(H, W)$ with RGB channels. We assume the videos across different views are temporally synchronized. A set of $N$ query points is defined by user, where each query point on view $v$ is represented by a 4-dimensional vector $q_v = (t_q, x_q, y_q)$. Here, $t_q$ denotes the frame index, and $(x_q, y_q)$ the spatial coordinates. The goal of this task is to predict a set of trajectories $\mathcal{T} \in \mathbb{R}^{V \times T \times N \times 2}$ and occlusion states $\mathcal{O} \in \mathbb{R}^{V \times T \times N \times 1}$ for the given queries, where $\mathcal{T}$ denotes the 2D pixel locations of $N$ points over $T$ time steps and $V$ views, and $\mathcal{O}$ indicates whether each point is visible or occluded across views and time. Notably, unlike single-view point tracking which often fail to estimate the position of a point during and after occlusion (Harley et al., 2022), the multi-view setting allows referencing complementary information from other visible views to maintain consistent tracking.

## 4 METHODOLOGY

In this section, we present our multi-view point tracking model, **MV-TAP** (Tracking Any Point in Multi-view Videos), which integrates cross-view and spatio-temporal information to enable robust point tracking in multi-view videos. The MV-TAP architecture builds upon the single-view 2D point tracking baseline (Karaev et al., 2024a), inheriting its knowledge of single-view tracking. We first

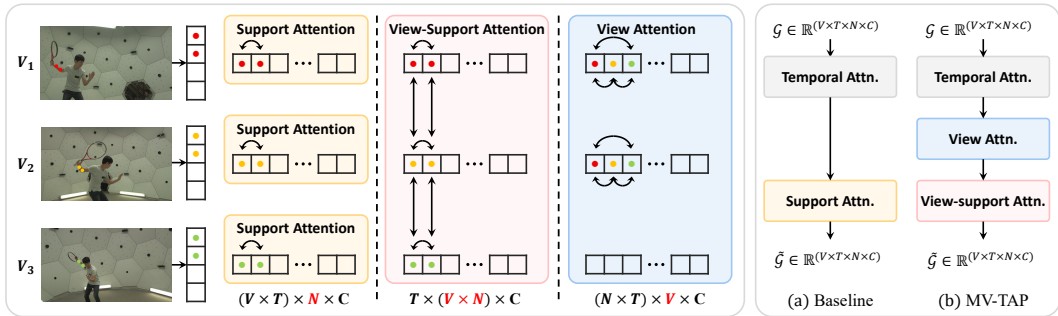

**Figure 3: Attention variants.** Within each view, temporal attention leverages temporal smoothness, and support attention exploits the co-movement of trajectories, following existing single-view point tracking methods (Karaev et al., 2024b). To extend this to the multi-view setting, we introduce two types of attention mechanisms: view-support attention and view attention. The former attends across all tracks from different views, while the latter aggregates information across views only for the same timestep and the tracks corresponding to a given query point.

describe the single-view point tracking pipeline, and then introduce the multi-view specific design that allows the model to effectively leverage the abundant information available in multiple views.

## 4.1 POINT TRACKING WITHIN A SINGLE-VIEW

Following recent work (Cho et al., 2024; Karaev et al., 2024a) in single-view point tracking, we refine initial correspondences using a local 4D correlation volume. Given an initial track hypothesis $\mathcal{T}^0$, which can be initialized either by a constant trajectory (Karaev et al., 2024b) or by feature matching (Doersch et al., 2023), we define the local correlation around a query coordinate $q = (t_q, x_q, y_q)$ and its hypothesized match $p = (t_p, x_p, y_p)$.

We first introduce the local neighborhoods around $p$ and $q$ as

$$\mathcal{N}(p, r_p) = \{\, p + \delta \mid \delta \in \mathbb{Z}^2, \|\delta\|_\infty \leq r_p \,\}, \quad \mathcal{N}(q, r_q) = \{\, q + \delta \mid \delta \in \mathbb{Z}^2, \|\delta\|_\infty \leq r_q \,\}, \quad (1)$$

where $r_p$ and $r_q$ denote the spatial radii. The local 4D correlation tensor is then defined as

$$\mathcal{L}_t(i, j; p, q) = \frac{F_t(i) \cdot F_{t_q}(j)}{\|F_t(i)\|_2 \, \|F_{t_q}(j)\|_2}, \quad i \in \mathcal{N}(p, r_p), \; j \in \mathcal{N}(q, r_q), \quad (2)$$

where $F_t$ and $F_{t_q}$ denote the feature maps from frame $t$ and the query frame $t_q$, respectively. The resulting tensor has dimensions $(2r_p + 1) \times (2r_p + 1) \times (2r_q + 1) \times (2r_q + 1)$, capturing all pairwise similarities between the two neighborhoods.

At each timestep, the correlation $\mathcal{L}_t$ and the current hypothesis position are encoded into a token. Stacking tokens across frames and query points forms an input tensor $X \in \mathbb{R}^{T \times N \times d}$, where $T$ is the number of frames, $N$ the number of query points, and $d$ the feature dimension. This sequence is processed by a Transformer that interleaves *temporal attention* and *support attention*, each applied along a specific axis while the feature dimension $d$ remains unchanged.

**Temporal Attention.** Temporal attention aggregates information along the **time axis** $T$. Formally, given query, key, and value projections $Q_T, K_T, V_T \in \mathbb{R}^{T \times d}$ for a fixed point,

$$\text{Attn}_{\text{temp}}(X) = \text{softmax}\left(\frac{Q_T K_T^\top}{\sqrt{d}}\right) V_T, \quad (3)$$

which integrates evidence across the frame sequence, ensuring temporally smooth trajectory updates.

**Support Attention.** Support attention aggregates information along the **point axis** $N$ within a single frame. Formally, given projections $Q_N, K_N, V_N \in \mathbb{R}^{N \times d}$ for a fixed timestep,

$$\text{Attn}_{\text{sup}}(X) = \text{softmax}\left(\frac{Q_N K_N^\top}{\sqrt{d}}\right) V_N, \quad (4)$$

which captures rigidity priors by linking points with consistent motion patterns.

In practice, temporal and support attention are interleaved across Transformer layers, so that temporal consistency and intra-view point coherence reinforce each other. Through iterative refinement, the model outputs the final single-view trajectory $\mathcal{T}$ and its occlusion status $\mathcal{O}$.

**Trajectory and Occlusion Updates.** Concretely, at each refinement step, the Transformer predicts incremental updates to both the track position and occlusion state:

$$\Delta\mathcal{T}, \ \Delta\mathcal{O} = \text{Transformer}(X). \tag{5}$$

These updates are applied to the previous estimates as

$$\mathcal{T}^{(m)} = \mathcal{T}^{(m-1)} + \Delta\mathcal{T}, \quad \mathcal{O}^{(m)} = \mathcal{O}^{(m-1)} + \Delta\mathcal{O}, \tag{6}$$

so that after $M$ refinement steps, the model produces the final trajectory $\mathcal{T}$ and occlusion status $\mathcal{O}$.

## 4.2 Extending the Transformer to Multi-View

While single-view trackers perform reliably under smooth motion, they often fail under extreme transformations such as large rotations or occlusions. Multi-view videos provide complementary perspectives, allowing consensus across views to resolve such ambiguities. To leverage this, we extend the single-view Transformer with additional modules that perform attention across views. We now describe two attention mechanisms that complement each other: view-support attention, and view attention.

In the multi-view setting, we construct an input token sequence for each view, following the same procedure as in single-view point tracking. This results in a sequence of shape $\mathbb{R}^{V \times T \times N \times d}$, where $V$ is the number of views, $T$ the number of frames, $N$ the number of query points, and $d$ the feature dimension. When applying attention across different axes, the feature dimension $d$ is always preserved, while the other axes are flattened into the batch dimension to enable attention along the selected axis, as illustrated in Figure 3.

**View-Support Attention.** We extend support attention to the multi-view case by applying attention along the **flattened view–point axis** $(V \cdot N)$. Formally, $Q_{V \cdot N}, K_{V \cdot N}, V_{V \cdot N} \in \mathbb{R}^{(V \cdot N) \times d}$,

$$\text{Attn}_{\text{vs}}(X) = \text{softmax}\left(\frac{Q_{V \cdot N} K_{V \cdot N}^{\top}}{\sqrt{d}}\right) V_{V \cdot N}. \tag{7}$$

This enables joint reasoning about intra-view and inter-view relations. To reduce computational cost, we adopt proxy tokens (Karaev et al., 2024a).

**View Attention.** Finally, to explicitly align representations across different views, we apply attention along the **view axis** $V$. Here, $Q_V, K_V, V_V \in \mathbb{R}^{V \times d}$,

$$\text{Attn}_{\text{view}}(X) = \text{softmax}\left(\frac{Q_V K_V^{\top}}{\sqrt{d}}\right) V_V. \tag{8}$$

This ensures cross-view consistency by directly linking tokens from different cameras.

With temporal, support, view-support, and view attention, the Transformer iteratively refines point trajectories and occlusion probabilities. In each refinement step, we alternate temporal, view, and view-support attention to jointly model cross-view spatio-temporal relationships. After $M$ refinement iterations, the model outputs the final multi-view trajectories $\mathcal{T}$ and occlusion states $\mathcal{O}$, as defined in our task formulation.

## 4.3 Training

To train MV-TAP, we optimize both trajectory regression and occlusion prediction. For trajectory supervision, we use the Huber loss (Huber, 1992), applied to both visible and occluded points with different weights, following Karaev et al. (2024a):

$$\mathcal{L}_{\text{track}}(T, \mathcal{T}^*) = \sum_{m=1}^{M} \gamma^{M-m} \left(\tfrac{I_{\text{occ}}}{5} + I_{\text{vis}}\right) \ell_{\text{Huber}}\left(\mathcal{T}^{(m)}, \mathcal{T}^*\right), \tag{9}$$

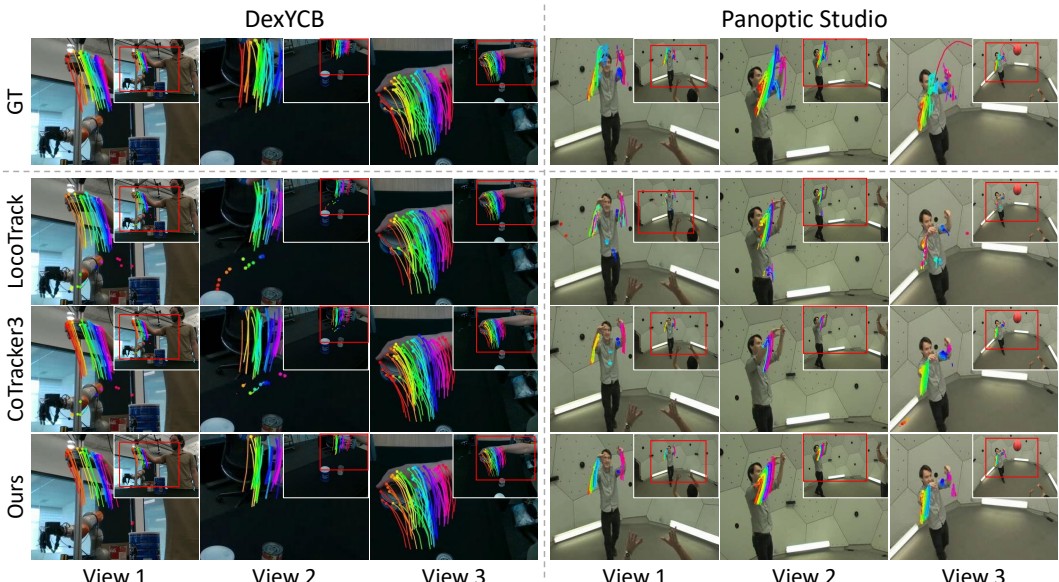

**Figure 4: Qualitative comparison.** Each example comprises three synchronized views of the same scene and each column shows one camera view. Compared to single-view baselines, MV-TAP demonstrates robust point tracking under occlusions and large motions, substantially reducing missing trajectories and untracked points.

where $\mathcal{T}^{(m)}$ is the predicted trajectory at refinement step $m$, $\mathcal{T}^*$ is the ground-truth trajectory, and $I_{\text{vis}}, I_{\text{occ}}$ indicate visibility or occlusion masks.

For occlusion supervision, we use a Binary Cross-Entropy (BCE) loss. We apply a sigmoid activation to the occlusion logits $\mathcal{O}^{(m)}$ before evaluating the loss:

$$\mathcal{L}_{\text{occ}}(\mathcal{O}, \mathcal{O}^*) = \sum_{m=1}^{M} \gamma^{M-m} \, \text{BCE}\Big( \sigma(\mathcal{O}^{(m)}), \ \mathcal{O}^* \Big), \qquad (10)$$

where $\hat{\mathcal{O}}^{(m)}$ is the predicted occlusion state at refinement step $m$ and $\mathcal{O}^*$ is the ground truth.

## 5 EXPERIMENTS

### 5.1 EXPERIMENTAL SETUP

**Training.** Since existing training datasets for point tracking are available for only the single-view scenarios, we generate a synthetic dataset for the multi-view point tracking task using Kubric engine (Greff et al., 2022). Our dataset, MV-TAP-Kub, consists of synchronized multi-view videos for 5,000 scenes, along with annotations that include point trajectories and their corresponding occlusion states. We provide further details of MV-TAP-Kub in Appendix B.

Our model is trained on the generated multi-view dataset for 50K steps on 8 NVIDIA A6000 GPUs with a batch size of 1 per GPU. We employ AdamW optimizer (Loshchilov & Hutter, 2017) with a learning rate of $10^{-4}$ and a weight decay of $10^{-4}$. We utilize a cosine learning rate scheduler with a 1,000 step warm-up stage and apply gradient clipping with a threshold of 1.0 for stable convergence.

As MV-TAP is based on CoTracker3 (Karaev et al., 2024a), we adopt the pretrained weights to initialize the feature encoder and transformer layers. During training, we update only the attention layers, while keeping all other pretrained parameters frozen. The number of input views is randomly selected between 1 and 4, the input resolution is $256 \times 256$, number of trajectories is 256. We set the number of refinement iterations to $M = 4$, and the spatial radii for local 4D correlation to $r_p = r_q = 3$.

**Table 1: Quantitative Comparison.** MV-TAP, trained on our MV-TAP-Kub dataset, achieves significant improvements over single-view tracking baselines. Our approach consistently outperforms DexYCB (Chao et al., 2021) and Panoptic Studio (Joo et al., 2015) with both view sampling modes, demonstrating the advantages of multi-view based tracking.

| Method | Training Dataset | DexYCB | | | Panoptic Studio (nearest) | | | Panoptic Studio (random) | | |
|---|---|---|---|---|---|---|---|---|---|---|
| | | AJ | $< \delta_{avg}^x$ | OA | AJ | $< \delta_{avg}^x$ | OA | AJ | $< \delta_{avg}^x$ | OA |
| TAPIR (Doersch et al., 2023) | Kub | 31.7 | 49.5 | 73.1 | 27.4 | 42.9 | 66.2 | 26.2 | 41.3 | 66.0 |
| CoTracker2 (Karaev et al., 2024b) | Kub | 28.2 | 52.2 | 64.0 | 38.1 | 59.8 | 71.8 | 30.3 | 51.3 | 69.3 |
| LocoTrack (Cho et al., 2024) | Kub | 36.6 | 57.6 | 72.0 | 41.7 | 60.8 | 72.3 | 36.7 | 56.1 | 69.1 |
| CoTracker3 (Karaev et al., 2024a) | Kub | 38.2 | 58.2 | 71.9 | 45.5 | 64.2 | 76.2 | 38.4 | 58.8 | 70.0 |
| **MV-TAP** | Kub + MV-TAP-Kub | 46.1 | 59.2 | 78.6 | 46.6 | 66.3 | 77.3 | 39.2 | 60.8 | 68.6 |

**Table 2: Ablation on different multi-view attention variants**. Adding view-support and view-aware attention improves performance over the baseline (Karaev et al., 2024a). MV-TAP, which combines both, achieves the best performance across all benchmarks.

| Method | DexYCB | | | Panoptic Studio(nearest) | | | Panoptic Studio(random) | | |
|---|---|---|---|---|---|---|---|---|---|
| | AJ | $< \delta_{avg}^x$ | OA | AJ | $< \delta_{avg}^x$ | OA | AJ | $< \delta_{avg}^x$ | OA |
| Baseline | 38.2 | 58.2 | 71.9 | 45.5 | 64.2 | 76.2 | 38.4 | 58.8 | 70.0 |
| Baseline + view-support attn. | 41.0 | 57.1 | 74.6 | 45.4 | 64.6 | 76.0 | 37.4 | 59.9 | 67.2 |
| Baseline + view attn. | 44.6 | **59.9** | 76.8 | 46.3 | 66.2 | **77.5** | 36.6 | 57.5 | **71.0** |
| **MV-TAP** | **46.1** | 59.2 | **78.6** | **46.6** | **66.3** | 77.3 | **39.2** | **60.8** | 68.6 |

**Table 3: Impact of the number of views**. We compare MV-TAP with the baseline (Karaev et al., 2024b) using a varying number of input views. Notably, occlusion accuracy improves significantly as the number of views increases.

| Method | 2 views | | | 4 views | | | 6 views | | | 8 views | | |
|---|---|---|---|---|---|---|---|---|---|---|---|---|
| | AJ | $< \delta_{avg}^x$ | OA | AJ | $< \delta_{avg}^x$ | OA | AJ | $< \delta_{avg}^x$ | OA | AJ | $< \delta_{avg}^x$ | OA |
| Baseline | 48.1 | 65.1 | 78.9 | 48.1 | 63.8 | 80.0 | 36.7 | 55.2 | 72.1 | 38.2 | 58.2 | 71.9 |
| **MV-TAP** | **51.8**(+3.7) | **67.6**(+2.5) | **81.8**(+2.9) | **51.9**(+3.8) | **66.6**(+2.8) | **82.6**(+2.6) | **42.7**(+6.0) | **57.4**(+2.2) | **77.5**(+3.4) | **46.1**(+7.9) | **59.2**(+1.0) | **78.6**(+6.7) |

**Evaluation Protocol.** We evaluate our method and baselines on the DexYCB hand dexterity dataset (Chao et al., 2021) and the Panoptic Studio dataset (Joo et al., 2015). For these benchmarks, we utilize the ground-truth point tracking annotations provided by Koppula et al. (2024) and Chao et al. (2021), respectively. For a consistent evaluation, we sample a fixed number of the available views from the DexYCB (8 views) and Panoptic Studio (31 views) datasets. We conduct evaluations in two sampling modes: nearest and random mode. The nearest mode selects geometrically close to a reference view, while the random mode selects views randomly from the entire available set. We assume that initial query points corresponding across all sampled views are given.

We use the standard point tracking metrics from TAP-Vid (Doersch et al., 2022), including position accuracy ($< \delta_{avg}^x$), occlusion accuracy (OA), and Average Jaccard (AJ). $< \delta_{avg}^x$ represents the average Percentage of Correct Keypoints (PCK) to evaluate the accuracy of predicted keypoint position. Concretely, it is computed by averaging PCK over error thresholds of 1, 2, 4, 8, and 16 pixels for visible points in ground-truth. OA denotes the accuracy of the binary prediction for occlusion. AJ is a composite score that jointly evaluates position and occlusion prediction of each point.

**Baselines.** We compare our method against recent state-of-the-art point tracking methods, including TAPIR (Doersch et al., 2023), CoTracker (Karaev et al., 2024b), LocoTrack (Cho et al., 2024), and CoTracker3 (Karaev et al., 2024a). For these single-view tracking baselines, which are designed for a monocular video setting, we perform tracking independently on each view.

## 5.2 MAIN RESULTS

**Quantitative Results.** We compare our method with recent state-of-the art single-view point trackers (Doersch et al., 2023; Karaev et al., 2024b;a; Cho et al., 2024) on DexYCB (Chao et al., 2021) and Panoptic Studio (Joo et al., 2015). For this comparison, we use 8 views for each benchmark; all 8 views are used for DexYCB, while for Panoptic Studio, they are sampled in two modes. As shown in Table 1, MV-TAP outperforms consistently on both benchmarks. In DexYCB, MV-TAP achieved a +7.9 AJ and +6.7 OA improvements while the hand-object interaction dataset has frequent occlusion state flipping. In Panoptic Studio, MV-TAP also achieves improvements across different view sampling methods, showing robustness in complex multi-view scenarios.

**Table 4: Comparison of feature-matching based query initialization across different backbones.** Results are reported with both soft-argmax and hard-argmax localization on Panoptic Studio and DexYCB. DINOv3 exhibits consistent performance across datasets and localization strategies, while VGGT achieves the highest accuracy on DexYCB. We select the four nearest views, and randomly sample a view and frame in which the point is visible for every query. The recovered locations using Algorithm 1 are subsequently compared against ground-truth annotations to quantify performance. Note that ResDINO and VGGT-DINO are SD-DINO-like (Zhang et al., 2023) variants designed to adapt the DINOv3 feature representation to different backbone architectures.

| Method | DexYCB | | Panoptic Studio | |
|---|---|---|---|---|
| | Soft Argmax($< \delta_{avg}^x$) | Argmax($< \delta_{avg}^x$) | Soft Argmax($< \delta_{avg}^x$) | Argmax($< \delta_{avg}^x$) |
| ResNet (He et al., 2016; Karaev et al., 2024a) | 12.0 | 28.0 | 29.7 | 68.1 |
| ResDINO | 31.0 | 35.4 | **57.7** | 62.9 |
| VGGT (Wang et al., 2025) | **53.9** | **52.6** | 36.0 | 41.0 |
| VGGTDINO | 24.5 | 24.1 | 22.3 | 25.7 |
| DINOv3 (Siméoni et al., 2025) | 36.9 | 48.0 | 54.6 | **75.7** |

**Table 5: Multi-view tracking with feature-matching queries.** On DexYCB, queries are initialized by feature matching with DINOv3 and tracked across views. For the baseline (Karaev et al., 2024a), the same queries are processed independently by a single-view model. MV-TAP shows clear improvements in AJ, $\delta_{avg}^x$, and OA.

| Method | DexYCB | | |
|---|---|---|---|
| | AJ | $< \delta_{avg}^x$ | OA |
| Baseline + DINOv3 | 35.1 | 48.5 | 80.8 |
| **MV-TAP** + DINOv3 | **37.7** (+2.6) | **50.4** (+1.9) | **84.9** (+4.1) |

**Table 6: Comparison under Farthest View Sampling.** MV-TAP shows superior performance compared to the baseline (Karaev et al., 2024a) under farthest view sampling, where views are sampled to maximize the distances between cameras.

| Method | DexYCB | | |
|---|---|---|---|
| | AJ | $< \delta_{avg}^x$ | OA |
| Baseline | 31.6 | 51.6 | 69.2 |
| **MV-TAP** | **36.6** (+5.0) | **52.2** (+0.6) | **72.2** (+3.0) |

**Table 7: Can multi-view information help point tracking?** For queries defined in a specific view, the baseline (Karaev et al., 2024a) tracks points within that view only, while MV-TAP augments each query by localizing correspondences across other views and tracks them jointly in a multi-view setting. We further analyze augmented query quality: using all queries, filtering errors > 16 px, and 8 px. Results on DexYCB and Panoptic Studio show that multi-view queries generally improve tracking, with larger gains under stricter filtering.

| Method | DexYCB | | | Panoptic Studio | | |
|---|---|---|---|---|---|---|
| | AJ | $< \delta_{avg}^x$ | OA | AJ | $< \delta_{avg}^x$ | OA |
| Baseline | 45.7 | 63.2 | 76.7 | **54.8** | **73.5** | **81.7** |
| **MV-TAP** | **47.1** (+1.4) | **63.7** (+0.5) | **78.1** (+1.4) | 54.2 | 73.4 | 79.6 |
| Baseline ($< \delta^{16}$) | 51.3 | 66.2 | 78.0 | **56.7** | **75.1** | **83.0** |
| **MV-TAP** ($< \delta^{16}$) | **56.9** (+5.6) | **69.7** (+3.5) | **84.7** (+6.7) | 56.3 | 74.8 | 80.7 |
| Baseline ($< \delta^{8}$) | 51.9 | 66.5 | 79.8 | **58.9** | **76.3** | 82.8 |
| **MV-TAP** ($< \delta^{8}$) | **55.6** (+3.7) | **70.0** (+3.5) | **82.8** (+3.0) | 58.5 | 76.0 | **82.9** (+0.1) |

**Qualitative Results.** We present qualitative comparison in Figure 4. We visualize the results from the DexYCB and Panoptic Studio. MV-TAP shows the superior robustness to large motions, demonstrating the effectiveness of multi-view information for point tracking.

## 5.3 ABLATION AND ANALYSIS

**Ablation on Attention Variants.** In Table 2, we present an ablation study on the attention architecture. We examine the effect of view-support attention, view attention, and ours which both view-support and view attentions are applied. These results show that both view-support and view attention individually improve performance over the baseline. Specifically, view-support attention aggregates all points across views into a shared space, improving robustness to occlusion and viewpoint variations. The view attention exchanges information across views for the points at the same timestep, enhancing multi-view consistency. Notably, the combined version, which view-support and view attention are applied, achieves the best performance, demonstrating their complementary benefits.

**Ablation on The Number of Views.** Although MV-TAP is trained with 1 to 4 views due to resource limitations, owing to the attention mechanism, it can handle an arbitrary number of views even larger than 4. We ablate the effect of number of views on DexYCB dataset with the nearest view sampling mode. As shown in Table 3, the performance of baseline degrades as the number of views increases, since more challenging views are introduced. In contrast, MV-TAP shows consistent improvements, demonstrating the ability to leverage the additional view information.

**Tracking with Feature Matching Mode.** In this experiment, we assume the query point is provided only in a single view and investigate the quality of query initialization in other views. To this end, we perform simple feature matching to localize the query points in additional views using five different backbones: ResNet (He et al., 2016), DINOv3 (Siméoni et al., 2025), VGGT (Wang et al., 2025), and two SD-DINO (Zhang et al., 2023)-like variants, ResDINO and VGGTDINO. For each backbone, we compute the dot-product between the query point feature and corresponding features in other views, and localize the query point by applying either soft-argmax or hard-argmax operations. As shown in Table 4, DINOv3 with hard-argmax operation achieves the best performance.

Based on this feature matching initialization, we integrate the matched points into our tracking pipeline and then evaluate tracking accuracy. Notably, when points are filtered by comparing between ground-truth position and predicted points by DINOv3 with the hard-argmax operation for rejecting erroneous tracks. In Table 5, we present a quantitative comparison of our method against the baseline using a DINOv3-based feature-matching initialization. Our method achieves more robust performance by refining the noise introduced during feature matching.

**Can MV-TAP Enhance Robustness in Complex Setting?** We investigate this question by evaluating on the DexYCB dataset with 4 sampled viewpoints which are sampled by farthest view sampling. In Table 6, MV-TAP demonstrates robustness under the random viewpoints compared to the baseline. This improvement is especially evident in AJ $(+5.0)$, represents more reliable predictions under challenging scenarios.

**Can Multi-view Information Help Point Tracking?** We examine whether multi-view tracking benefits from cross-view information. Unlike previous settings where query points from the ground-truth are provided in all views, we provide a query only in a single view. The corresponding points in other views are then obtained by feature matching as described in Appendix A. Since such matching can introduce noises, we compare MV-TAP against the single-view baseline to assess whether leveraging multi-view information can mitigate such errors. To further analyze the effect of matched query quality, we progressively filter out noisy matches by discarding those that deviate from the ground truth by more than 16 or 8 pixels. We then evaluate tracking under these thresholds. As summarized in Table 7, MV-TAP improves over the baseline on DexYCB even when queries are noisy, and achieves larger gains as the quality of the queries increases, while on Panoptic Studio the performance remains comparable to the baseline.

## 6 CONCLUSION

This work establishes multi-view 2D point tracking as a new and important task for advancing reliable spatio-temporal correspondence in dynamic, real-world scenes. By introducing MV-TAP, a framework that aggregates cross-view information through attention, we demonstrate how leveraging uncalibrated multi-view inputs can overcome key limitations of monocular trackers such as occlusion and motion ambiguity. Together with a large-scale synthetic dataset specifically designed for this task, our contributions provide both a principled formulation of the problem and a strong baseline method, paving the way for future research in robust multi-view point tracking.

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

# APPENDIX

## A  FEATURE MATCHING

When the query is given in only a single view, the query location in each other view is determined through comparing the query feature to feature maps of other views. Given a query point $q = (x_q, y_q, t_q, v_q)$, we extract its feature vector $\mathbf{f}_q$ from the backbone feature map $\Phi(V_{v_q, t_q})$ at $(x_q, y_q)$. For each candidate view $v \neq v_q$ and frame $t$, we compute a correlation map by dot product as

$$C_{v,t}(x, y) = \langle \mathbf{f}_q, \mathbf{F}_{v,t}(x, y) \rangle, \tag{11}$$

where $\mathbf{F}_{v,t} = \Phi(V_{v,t})$. We then select the most relevant frame for each view by

$$t_v^* = \arg \max_t \max_{x,y} C_{v,t}(x, y), \tag{12}$$

and localize the 2D correspondence using either hard-argmax

$$(x_v^*, y_v^*) = \arg \max_{x,y} C_{v,t_v^*}(x, y), \tag{13}$$

or soft-argmax

$$(x_v^*, y_v^*) = \sum_{x,y} (x, y) \cdot \frac{\exp(C_{v,t_v^*}(x, y))}{\sum_{x',y'} \exp(C_{v,t_v^*}(x', y'))}. \tag{14}$$

The hard-argmax provides discrete localization, while the soft-argmax yields a differentiable approximation more suitable for training. The resulting set of correspondences $\{(x_v^*, y_v^*, t_v^*, v)\}_{v \neq v_q}$ is then used as the initialization for cross-view tracking, and the performance of this initialization is strongly influenced by the choice of backbone and matching strategy.

### QUALITATIVE ANALYSIS OF FEATURE MATCHING

We provide qualitative visualizations of the correlation maps obtained by different backbones in Figure 5. Given a reference frame and a query point (marked by a red dot), we observe that DINOv3 (Siméoni et al., 2025) produces sharper and more localized activation maps compared to other feature extractors, leading to more reliable correspondences. In contrast, ResNet (He et al., 2016), whose parameters are taken from CoTracker3 (Karaev et al., 2024a), tends to generate noisy responses with weak localization, while ResDINO and VGGT (Wang et al., 2025) sometimes highlight semantically related but spatially inaccurate regions. VGGT-DINO shows intermediate behavior, capturing broader structures but lacking precise localization. Note that ResDINO and VGGT-DINO are SD-DINO-like variants (Zhang et al., 2023), adapting DINOv3 features to different backbone architectures. These comparisons suggest that the choice of backbone significantly impacts query initialization quality, with DINOv3 offering the most consistent and discriminative correspondences across diverse scenes.

## B  TRAINING DATASET DETAILS

To the best of our knowledge, there exists no large-scale dataset for training point tracking in a multi-view setting. For this reason, we generate a synthetic dataset using the Kubric engine (Greff et al., 2022). Specifically, we capture video from four views for each scene. To ensure that a sufficient number of points are visible across all views, we project points from every view to all other views. To enable the model to capture correlations across the views effectively, we sample the camera positions in chained manner. Each camera position is sampled within a certain angular range from a randomly selected previous view, without overlapping any already sampled views. Consequently, our training dataset comprises 4 synchronized multi-view videos for each of 5,000 dynamic scenes, with 132,608 annotated point trajectories including occlusion status.

## C  ADDITIONAL ABLATION AND ANALYSIS

While Table 3 focuses on MV-TAP and the baseline, Table 8 extends the analysis to single-view trackers and the proposed attention variants. We observe that most of single-view trackers suffer from significant performance degradation as the number of views increases. In contrast, the attention variants and MV-TAP show robustness to the increasing number of view

---

**Algorithm 1** Feature Matching Initialization

---

**Require:** Multi-view videos $V$, query point $q = (x_q, y_q, t_q, v_q)$
**Ensure:** Cross-view correspondences $\{(x_v^*, y_v^*, t_v^*, v)\}_{v \neq v_q}$

    Extract query feature $\mathbf{f}_q \leftarrow \Phi(V_{v_q, t_q})(x_q, y_q)$
    **for** each view $v \neq v_q$ **do**
        **for** each frame $t$ **do**
            Compute correlation map

$$C_{v,t}(x, y) \leftarrow \langle \mathbf{f}_q, \mathbf{F}_{v,t}(x, y) \rangle$$

        **end for**
        Select relevant frame

$$t_v^* \leftarrow \arg\max_t \max_{x,y} C_{v,t}(x, y)$$

        Localize correspondence by argmax

$$(x_v^*, y_v^*) \leftarrow \arg\max_{x,y} C_{v,t_v^*}(x, y)$$

        **end for**
    **return** $\{x_v^*, y_v^*, t_v^*\}$

---

**Table 8: Additional multi-view ablation with various single-view point tracking**.

| Method | 2 views | | | 4 views | | | 6 views | | | 8 views | | |
|---|---|---|---|---|---|---|---|---|---|---|---|---|
| | AJ | $< \delta_{avg}^x$ | OA | AJ | $< \delta_{avg}^x$ | OA | AJ | $< \delta_{avg}^x$ | OA | AJ | $< \delta_{avg}^x$ | OA |
| TAPIR | 35.3 | 55.0 | 75.3 | 38.4 | 55.0 | 75.6 | 30.2 | 48.1 | 71.3 | 31.7 | 49.5 | 73.1 |
| CoTracker2 | 39.1 | 63.2 | 73.9 | 38.6 | 61.8 | 73.6 | 27.8 | 51.2 | 65.8 | 28.6 | 52.2 | 64.0 |
| LocoTrack | 47.0 | 66.6 | 79.7 | 47.6 | 66.3 | 80.7 | 34.9 | 56.1 | 71.8 | 36.6 | 57.6 | 72.0 |
| CoTracker3 | 48.1 | 65.1 | 78.9 | 48.1 | 63.8 | 80.0 | 36.7 | 55.2 | 72.1 | 38.2 | 58.2 | 71.9 |
| CoTracker3 + view-support attn. | 51.2 | 66.9 | 81.7 | 50.3 | 65.9 | 80.5 | 35.1 | 52.0 | 70.8 | 41.0 | 57.1 | 74.6 |
| CoTracker3 + view attn. | **52.9** | **68.2** | **82.4** | **52.9** | **67.6** | 82.2 | 40.1 | 57.1 | 74.4 | 44.6 | **59.9** | 76.8 |
| **MV-TAP** | 51.8 | 67.6 | 81.8 | 51.9 | 66.6 | **82.6** | **42.7** | **57.4** | **77.5** | **46.1** | 59.2 | **78.6** |

# D  ADDITIONAL VISUALIZATION

We provide additional qualitative results in Figure 6. These visualizations compare our method with baseline approaches across multiple views and illustrate the different motion patterns present in the dataset.

# E  REPRODUCIBILITY STATEMENT

We confirm that all results reported in this paper are fully reproducible. The experimental settings, datasets, preprocessing steps, and evaluation metrics are described in detail in the main text and Appendix. We will release the complete source code and pretrained models upon publication to facilitate independent verification of our results.

# F  THE USAGE OF LLM

We acknowledge the use of a large language model to assist with the writing of this paper. The model was used to help draft and edit portions of the text, including improving clarity, grammar, and overall readability. All technical content, experimental design, analysis, and conclusions were created and verified by the authors. The authors take full responsibility for the final contents of this submission.

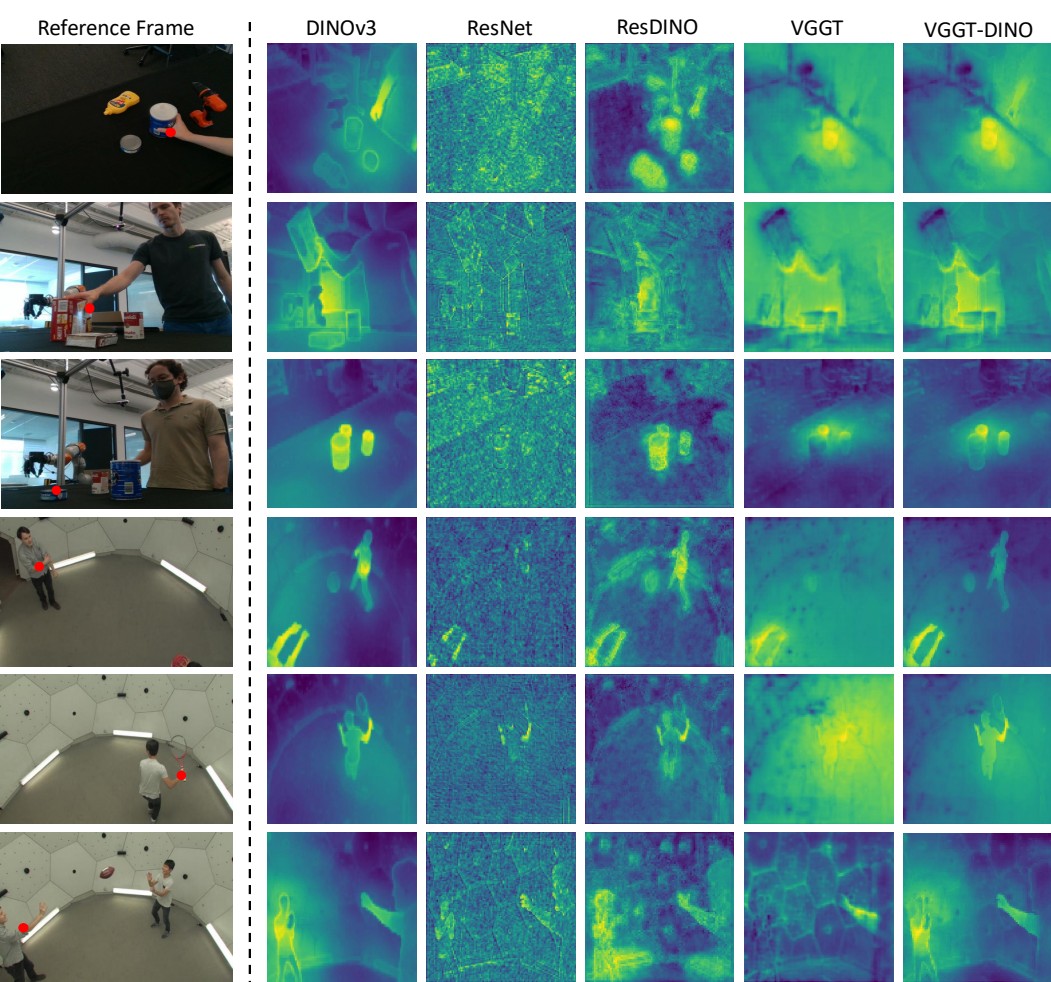

**Figure 5: Qualitative comparison of feature matching with different backbones.** We visualize the correlation maps obtained by DINOv3, ResNet, ResDINO, VGGT, and VGGT-DINO given a reference frame (left) and query point (red dot). The correlation maps are computed on images from a different target view, illustrating how each backbone localizes the query across views.

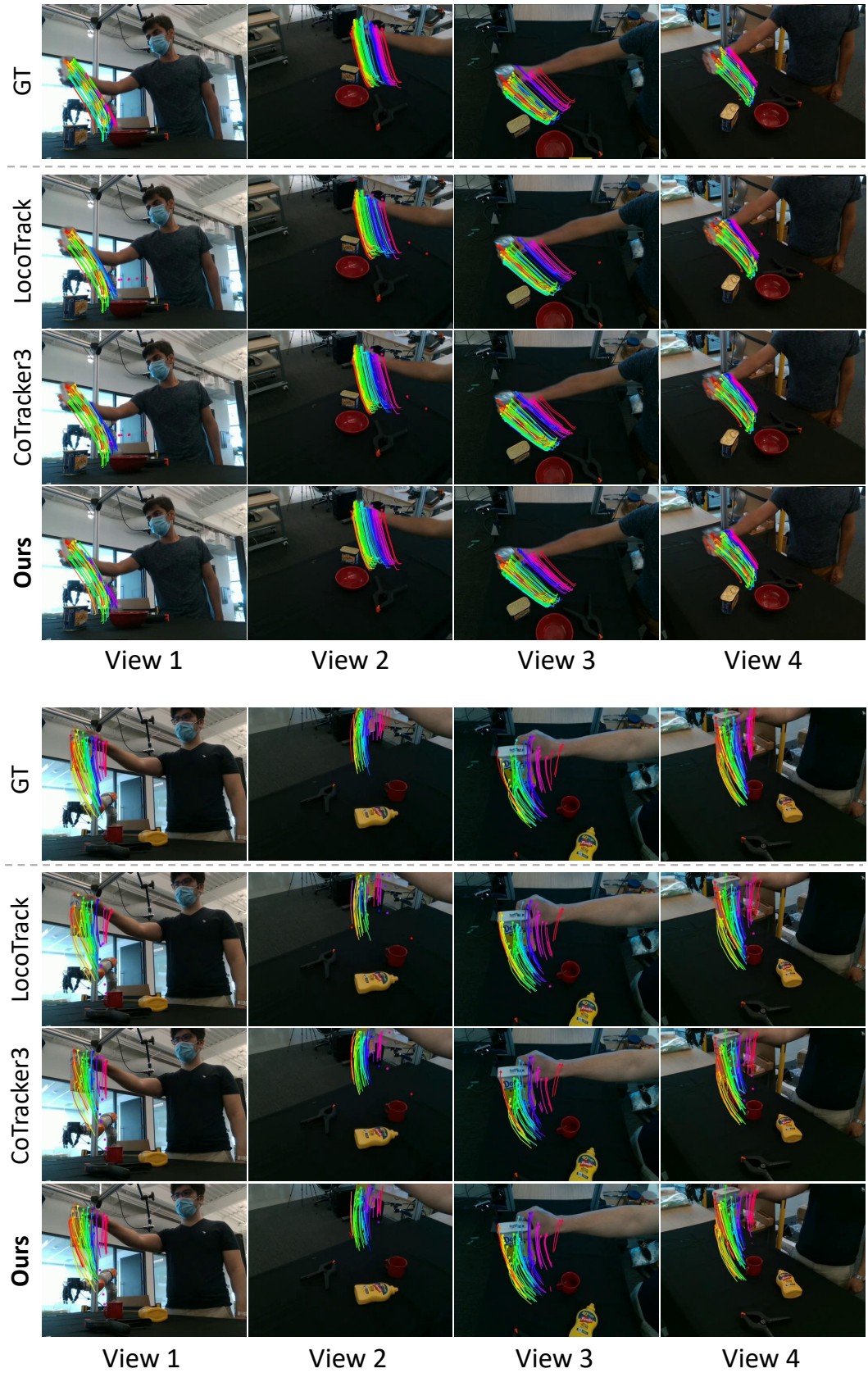

**Figure 6: Additional visualization of tracks from our pipeline and baselines.** In these examples, our method produces more accurate and temporally consistent multi-view tracks compared to prior approaches.

