# OpenReview forum: "Tracking Any Point In Multi-View Videos"
_ICLR.cc/2026/Conference — ICLR 2026 Conference Withdrawn Submission_

### Official Review · Reviewer_88NE · 2025-10-21

**Soundness:** 3
**Presentation:** 3
**Contribution:** 2
**Rating:** 4
**Confidence:** 4

**Summary:**

The authors propose a novel multi-view point tracking framework (MV-TAP) that leverages cross-frame information to enhance tracking quality. The framework introduces view-support and view-attention mechanisms to better aggregate information across views.

**Strengths:**

- The paper is well-written and easy to follow, with extensive experiments.

- The authors introduce a novel and meaningful problem: multi-view point tracking.

- The proposed view-support and view-attention mechanisms are interesting and technically sound.

**Weaknesses:**

1) I would like to discuss the necessity of the proposed problem setting. It is evident that utilizing multi-view information can improve tracking performance. However, in certain applications such as autonomous driving, the relative poses among multiple cameras are usually known, making multi-view fusion more straightforward. In contrast, in surveillance systems, multiple cameras often have little overlap, which limits the benefit of cross-view information. Additionally, there exist approaches such as MegaSAM and Monst3r that can reconstruct dynamic 3D scenes and directly perform 3D point tracking (as shown in TAPIP3D). It would be helpful if the authors could further clarify the real-world scenarios where multi-view point tracking provides distinct advantages.

2) Table 1: The comparison may not be entirely fair. The Kubric dataset contains around 10,000 videos, while MV-TAP-Kub has approximately 5,000 multi-view videos. The observed improvement could partly stem from the larger or more diverse training data. To make the comparison more convincing, it would be beneficial to train the baseline methods on MV-TAP-Kub as well (treating each view as a separate training sample).

3) Table 7: The results of MV-TAP on Panoptic Studio are consistently lower (or similar) compared to the baseline. Could the authors provide insights into why this occurs?

4) Tables 3 & 8: The results appear to decrease as the number of views increases. Could the authors discuss possible reasons behind this trend? Intuitively, tracking with more views should improve overall performance—some explanation for this counterintuitive behavior would strengthen the paper.

5) It would be helpful to clarify how the evaluation metrics are defined in the multi-view setting. Are the metrics computed separately for each view and then averaged?

6) (Suggestion) For initializing query points across different views, it may be more robust to use established feature-matching methods such as LoFTR, Dust3r, or Mast3r, instead of relying on simple argmax-based feature matching.

**Questions:**

Please check the weaknesses section

---

### Official Review · Reviewer_yH5X · 2025-10-25

**Soundness:** 2
**Presentation:** 3
**Contribution:** 2
**Rating:** 4
**Confidence:** 5

**Summary:**

The paper introduces MV-TAP, a transformer-based architecture for multi-view point tracking, aims to establish 2D spatio-temporal correspondences across synchronized, uncalibrated video streams. The proposed method extends the CoTracker3 single-view transformer via cross-view attention modules (view-support and view attention), enabling the integration of multi-view cues to improve robustness under occlusions and large motions. The authors also present a synthetic dataset, MV-TAP-Kub, generated using Kubric to train and evaluate the approach. Experiments on DexYCB and Panoptic Studio show moderate improvements over single-view baselines and detailed ablations on attention variants and number of views.

**Strengths:**

- Multi-view point tracking is an increasingly relevant research direction for correspondence modeling, particularly with applications in dynamic 4D reconstruction and geometric reasoning.
- The introduction of a synthetic Kubric-based dataset for multi-view tracking is valuable for future research reproducibility.
- The comprehensive ablations address natural questions that arise from reading the paper.

**Weaknesses:**

## Fairness of comparisons
- The comparison between multi-view MV-TAP and single-view baselines is not fair by design. The multi-view setup leverages significantly more information (8 synchronized views), while all baselines operate independently on single-view sequences. Apart from that, the meaningful comparison is only to the same backbone (CoTracker3) used as initialization, as it acts as a lower bound. The improvements over CoTracker3 are not substantial despite the multi-view input, raising questions about the true benefit of the current cross-view aggregation approach.

## Baselines and related work
- The baseline set is incomplete. No multi-view or 3D trackers (MVTracker [1], or geometry-aware 3D correspondence models such as TAPIP-3D [2]) are included, though they directly address similar objectives.
- The claim of being "the first to define multi-view point tracking" is inaccurate, as MVTracker already formulate similar multi-view correspondence problem.
- Running monocular trackers independently per view does not constitute a strong baseline; at least one multi-view fusion baseline (suc as simple 3D triangulation + reprojection) should be provided for context.

## Technical depth and novelty
- The architectural novelty is limited. The model adds two additional attention axes to an existing transformer but does not fundamentally change the reasoning process.

## Evaluation setup
- It is unclear whether the single-view baselines were evaluated per view and averaged or whether the authors aggregated across views. Averaging over view-specific performance could be misleading given viewpoint redundancy.

## Minor Comments
- Line 148: The query representation $q_v = (t_q, x_q, y_q)$ should be referred to as a 3-dimensional vector, not 4-dimensional.
- Better visual alignment between tables and text would improve readability in the later pages.
- Some phrasing ("first to define") should be moderated given existing literature.

---

## Overall
While the paper tackles an emerging and meaningful topic, extending point tracking into a multi-view setting, its conceptual novelty and fairness of comparison are limited. The contribution mainly adapts CoTracker3’s architecture with additional attention along the view axis, rather than introducing a fundamentally new paradigm. The quantitative gains over the single-view backbone are modest, and important multi-view baselines are missing.

---


## References
[1] Rajic et al., Multi-View 3D Point Tracking, ICCV 2025

[2] Zhang et al., TAPIP3D: Tracking Any Point in Persistent 3D Geometry, NeurIPS 2025

**Questions:**

-  Does training with multi-view inputs improve the single-view tracking capability of the model, i.e. generalization back to monocular setups?

---

### Official Review · Reviewer_GeH8 · 2025-10-26

**Soundness:** 2
**Presentation:** 3
**Contribution:** 2
**Rating:** 4
**Confidence:** 5

**Summary:**

This paper is clearly written and easy to follow. The authors’ motivation, to leverage complementary information from multiple views to enhance point tracking across views, is well stated and intuitive.

However, the overall novelty of the proposed setting is somewhat limited, and the problem formulation does not seem to align well with realistic application scenarios. Moreover, the current experimental results are not sufficient to convincingly support the motivation behind introducing this new setting.

**Strengths:**

The authors provide a clear and well-articulated motivation, and the dataset and baseline designed accordingly are well-defined and logically aligned with the stated goals.

**Weaknesses:**

1. The task of multi-view point tracking is not entirely new; similar settings have been explored in prior works [1], although they require per-scene optimization. The authors should include a discussion comparing their method and problem formulation to such approaches, clearly articulating the differences and advantages.

2. Practicality of the setting:

If my understanding is correct, the initial positions of the tracked points must be manually specified for each view. This assumption is unrealistic for practical applications. For instance, if a user wishes to track points across 32 views, they would need to manually identify and mark the corresponding locations for each point in all 32 views. Tracking 10 points would thus require specifying 320 initial positions—an extremely user-unfriendly setup. Furthermore, in real-world scenarios, a point is often not visible in all views simultaneously, making it even more challenging for users to accurately specify consistent initial positions.

3. Experimental support and analysis:

Although the paper claims that multi-view complementary information improves tracking in dynamic scenes, the empirical results only partially support this claim.

3.1 In Table 1, the improvement on Panoptic Studio is marginal.

3.2 While DexYCB shows a notable gain in the AJ metric, the improvement mainly stems from OA, whereas the tracking accuracy measure (\delta_x) shows only limited improvement.

3.3 Additionally, in Table 3, although MV-TAP appears to benefit from an increasing number of views, its absolute performance actually degrades as the view count grows, which contradicts the authors’ claim. I would recommend including an ablation study on Panoptic Studio similar to that in Table 3, to examine whether the observed advantages are consistent across datasets.

4. As a newly proposed setting, the paper should include more comparisons in the Quantitative Comparison section to verify the rationality of the setting and the superiority of the proposed method. If time permits, it is recommended to include more methods that perform well in dynamic scenarios (which can be selected based on their performance on the TAP-Vid-Kubric subset), such as the Track-On series, TAPTR series, and BootsTAP series.

[1] Wang L H, Cheng Y J, Liu T L. Tracking Everything Everywhere across Multiple Cameras[C]//Proceedings of the AAAI Conference on Artificial Intelligence. 2025, 39(7): 7789-7797.

**Questions:**

Please refer to Weaknesses.

---

### Official Review · Reviewer_PgXM · 2025-11-01

**Soundness:** 3
**Presentation:** 3
**Contribution:** 2
**Rating:** 4
**Confidence:** 4

**Summary:**

This paper introduces a multi-view point tracking approach for dynamic scenes captured by multiple uncalibrated videos. The author proposes a novel framework that leverages cross-view information to overcome the limitations of traditional single-view point tracking methods. To enhance performance, they design view attention and view-support attention module, and conduct extensive experiments to demonstrate the effectiveness of the proposed approach.

**Strengths:**

1. This paper defines the task of multi-view point tracking, which involves tracking query points across multiple uncalibrated videos of dynamic scenes.
2. The author designed view attention and view-support attention mechanisms and conducted extensive experiments to demonstrate the effectiveness of the proposed modules.

**Weaknesses:**

1. Training CoTracker3 only on the Kub dataset is not fair. The MV-TAP-Kub dataset includes 5,000 four-view videos, which can also be treated as 20,000 additional data samples.
2. This task requires accurately initialized query points that correspond across all sampled views, which may be impractical.
3. The description of the ablation study is confusing. I will include my questions in the Questions section to ensure that my understanding is correct.

**Questions:**

1. Can you train CoTracker3 with Kub and the separated MV-TAP-Kub dataset and present the results in Table 1?
2. I am also curious why you did not use support attention to replace view-support attention. Does view-support attention significantly outperform support attention?
3. In Table 7, can you provide results using ground-truth query points?
4. For Table 3, why does the baseline result change with the number of views? Did you only select several views from the DEXYCB dataset and not evaluate on the others? You should clarify this in the paper.
5. For Table 5, did you use the predicted initial query points (which may be inaccurate) to evaluate the single-view model?
6. For Table 7, why do the baseline results change? Did you discard the given query points whose predictions were incorrect?
7. I do not understand the difference between Table 5 and Table 7. Why do they show different baseline performances?

---

### Note · Authors · 2025-11-12

I have read and agree with the venue's withdrawal policy on behalf of myself and my co-authors.